# A Clinical Perspective on Plasma Cell Leukemia: A Single-Center Experience

**DOI:** 10.3390/cancers16112149

**Published:** 2024-06-05

**Authors:** Andrew Y. Li, Farin Kamangar, Noa G. Holtzman, Aaron P. Rapoport, Mehmet H. Kocoglu, Djordje Atanackovic, Ashraf Z. Badros

**Affiliations:** 1University of Maryland Greenebaum Comprehensive Cancer Center, Baltimore, MD 21201, USA; andrew.li@umm.edu (A.Y.L.); arapoport@umm.edu (A.P.R.); mkocoglu@umm.edu (M.H.K.); datanackovic@som.umaryland.edu (D.A.); 2Department of Biology, School of Computer, Mathematical, and Natural Sciences, Morgan State University, Baltimore, MD 21251, USA; farin.kamangar@morgan.edu; 3University of Miami Sylvester Comprehensive Cancer Center, Miami, FL 33136, USA; ngh45@med.miami.edu; 4Marlene & Stewart Greenebaum Comprehensive Cancer Center, University of Maryland School of Medicine, Baltimore, MD 21201, USA

**Keywords:** plasma cell leukemia, multiple myeloma, novel agents

## Abstract

**Simple Summary:**

We evaluated the clinical features of primary and secondary plasma cell leukemia and the impact of current therapies. High-risk cytogenetics, low platelets, extramedullary disease and high LDH were independently associated with a poor outcome, with an overall survival of 11% at 5 years.

**Abstract:**

Circulating plasma cells (CPCs) are detected in most multiple myeloma (MM) patients, both at diagnosis and on relapse. A small subset, plasma cell leukemia (PCL), represents a different biology and has a poor prognosis. In this retrospective analysis, we evaluated patients with primary (pPCL, n = 35) or secondary (sPCL, n = 49), with ≥5% CPCs and a smaller subset with lower CPCs of 1–4% (n = 20). The median age was 61 years; 45% were men and 54% were Black. High-risk cytogenetics were found in 87% and extramedullary disease in 47%. For the entire cohort, 75% received a proteasome inhibitor, 70% chemotherapy, 54% an immunomodulatory drug, 24% a daratumumab-based regimen and 26% an autologous stem cell transplant (ASCT). The treatments marginally improved the overall survival (OS) for pPCL vs. sPCL (13 vs. 3.5 months *p* = 0.002). However, the 5-year survival for the whole cohort was dismal at 11%. High-risk cytogenetics, low platelets, extramedullary disease and high LDH were independently associated with poor outcomes. Further research is urgently needed to expand the treatment options and improve the outcomes in PCL.

## 1. Introduction

Plasma cell leukemia (PCL) is a rare aggressive plasma cell dyscrasia that is divided into primary PCL (pPCL), in newly diagnosed patients, or secondary PCL (sPCL), in patients with a prior diagnosis of multiple myeloma (MM) [1]. The incidence of PCL ranges between 2 and 5%, as reported in mostly retrospective cohorts [2]. Today, with sensitive quantitative techniques, flow cytometry and PCR, most patients with MM are found to have low levels of circulating plasma cells (CPCs) [3,4]. Few studies addressed the correlation between MM, PCL and CPC and the optimal cut-off value to define PCL [5]. In 2021, the International Myeloma Working Group (IMWG) lowered the threshold to define pPCL from 20% with an absolute count of 2 × 10^9^/L CPCs to ≥5%, morphologically defined, based on two studies reporting similar outcomes [6,7,8,9]. Recent data show that sPCL patients have poor outcomes with ≥5% CPCs [10]. Less data are available for MM patients with 1–4% CPCs. In a recent study of 767 patients with newly diagnosed MM, the presence of ≥2% CPCs by morphology was 14%, with clinical features comparable to those of 33 pPCL cases [1]. Although the CPCs were not high enough to meet the diagnostic criteria for pPCL, the survival of MM patients with CPCs was comparable with that of pPCL, with a median progression-free survival (PFS) of 17 months and an overall survival (OS) of 25 months [11]. Similar data show that having ≥2% CPCs, defined by flow cytometry, carries a poor prognosis [12].

PCL is a highly proliferative stage in MM [13]. The prognosis is poor and survival varies from a few years in newly diagnosed pPCL to a few months in sPCL [14,15]. The clinical and biological features also vary; pPCL is associated with a lower age at diagnosis, extramedullary disease (EMD), higher creatinine and beta-2 microglobulin (B2M), hypodiploidy, t(11;14), and longer OS in comparison to sPCL, which tends to present with a higher age at diagnosis, bony involvement, lower platelet count, high serum M-protein, hyperdiploidy, and much shorter OS [16]. The optimal treatment regimens and sequencing are less well-defined for PCL as compared to MM. Despite the progress seen in MM, a parallel advancement has not been seen in PCL, in part due to the low incidence as well as the rarity of studies including PCL [17,18]. The goal of this large single-institution retrospective study is to describe the clinical presentation and treatment impact on overall survival in 104 patients with pPCL and sPCL (defined ≥5% CPC) as well as in patients with 1–4% CPCs.

## 2. Methods

### 2.1. Study Design

MM patients seen at a single center from 2004 to 2022 were included if CPCs were detected via peripheral blood manual differential (morphology) and/or flow cytometry. Patients were stratified by the percentage of CPCs into 3 groups: pPCL was defined as de novo PCL with ≥5% CPCs, sPCL was defined as PCL with ≥5% CPCs and prior diagnosis of MM, while intermediate (iCPC) had CPCs of 1–4%. Demographics, clinical data and survival outcomes were collected retrospectively. This study was approved by the Institutional Review Board and was conducted in accordance with the Declaration of Helsinki.

### 2.2. Statistical Analysis

Statistical analysis and graph generation were performed using Stata, version 17 (StataCorp, College Station, TX, USA). Pearson’s chi-squared test of independence was performed for categorical variables such as race, presence of lytic lesions on imaging, cytogenetic risk, etc. The mean differences between two groups (i.e., Black and non-Black) were also tested using independent sample *t*-tests. The mean differences between three or more groups (i.e., CPC/PCL type groups and WBC quartiles) were tested using one-way analysis of variance (ANOVA) followed by Scheffe tests. Numerical variables such as the white blood cell count (WBC) and lactate dehydrogenase (LDH) were divided into quartiles. The OS between CPC/PCL-type groups was compared using the log-rank test for the equality of survivor functions and Cox regression models after adjustment for the covariates of interest and utilizing the Breslow method for tied survival times. OS was graphed using the Kaplan–Meier method.

## 3. Results

A total of 104 patients were included, with 35 pPCL, 49 sPCL, and 20 iCPC. The median time from MM diagnosis to the development of sPCL was 30 months (range: 20–1364 months). There was no statistically significant difference between the degree of CPCs by flow cytometry and CPCs by peripheral blood manual differential (morphology) when data were available for comparison (*p* = 0.15). There were no differences in age (*p* = 0.82), gender (*p* = 0.87), or race (*p =* 0.21) between the groups. Overall, there were 56 (54%) Black patients, with more Black compared to non-Black females (68% vs. 40%; *p* = 0.0040) (Table 1).

Patients with PCL tended to have anemia, thrombocytopenia, abnormal renal function, and elevated LDH. No differences were noted between the groups regarding the B2M levels, serum M-spike, percentage of bone marrow plasmacytosis, or immunohistochemical expression of various markers. Patients with pPCL had a higher white blood cell count (*p* < 0.001) and calcium (*p* = 0.02) at the time of diagnosis than patients with sPCL and iCPC. Patients with pPCL were less likely to have lytic lesions on the positron emission tomography (PET/CT) scan compared to sPCL and iCPC patients (60% vs. 79% vs. 95%; *p* = 0.01). The incidence of extramedullary disease was high in the whole cohort at 47%; the incidence of lymphadenopathy and CNS involvement was higher in sPCL compared to pPCL (*p* = 0.02 and 0.03, respectively).

High-risk cytogenetics were reported in 87% of the entire cohort and defined according to the National Comprehensive Cancer Network (NCCN) guidelines as possessing any of the following: t(4;14), t(14;16), deletion 17p or *TP53* deletion/mutation, gain or amplification of 1q21, karyotypic deletion 13, or complex karyotype. iCPC was less likely to involve a complex karyotype compared to pPCL and sPCL (21% vs. 79% vs. 65%; *p* < 0.001). More patients with pPCL had hypodiploidy (*p* = 0.01). The pPCL patients were less likely to have deletion 17p compared to sPCL and iCPC (29% vs. 66% vs. 50%; *p* = 0.006); the high incidence (50%) in the iCPC group indicates more clones than seen in newly diagnosed MM (15%) [19]. There was no difference in the presence of high-risk cytogenetics between Black (85%) and non-Black (89%) patients (*p* = 0.53).

For the entire cohort, 75% received a proteasome inhibitor (bortezomib or carfilzomib), 70% cytotoxic chemotherapy (alkylating agents or DVT-PACE (dexamethasone-bortezomib-thalidomide-cisplatin-doxorubicin-cyclophosphamide-etoposide)), 54% immunomodulatory drugs (IMiDs: lenalidomide, pomalidomide, or thalidomide), 24% a daratumumab-based regimen, and 26% an autologous stem cell transplant (ASCT) (Figure 1). All the regimens except cytotoxic chemotherapy (HR 0.69, 95% CI 0.44–1.08; *p* = 0.11) and DVT-PACE (HR 0.97, 95% CI 0.62–1.50; *p* = 0.88) demonstrated an OS benefit in all the groups (Table 2). The lack of an OS benefit in the pPCL patients receiving DVT-PACE persisted on multivariate analysis when adjusting for the PCL group alone (HR 0.79, 95% CI 0.50–1.26; *p* = 0.34) and the PCL group, age, platelet count, and presence of high-risk cytogenetics (HR 1.11, 95% CI 0.63–1.92; *p* = 0.75); in pPCL, most patients who received DVT-PACE had failed or progressed after at least one line of induction, representing higher-risk patients with a higher tumor burden. The small number of patients (n = 17) who achieved a durable/stable response to upfront induction and proceeded to ASCT had an improved OS (HR 0.54, 95% CI 0.30–0.99; *p* = 0.05). Analysis was not performed for patients who received chimeric antigen receptor T-cell (CAR-T) therapy (n = 3) or allogeneic stem cell transplant (n = 6) due to the low numbers of patients who received these treatments as well as the lack of impact on the OS.

The median OS for the whole cohort was 7.14 months at a median follow-up of 6.7 months. When stratified by group, there was a statistically significant difference in the median OS between the groups, *p* = 0.02. The pPCL patients had a higher median OS versus sPCL (13.1 vs. 3.5 months, *p* = 0.002) but a similar OS to the iCPC cohort (13.1 vs. 10.0 months, *p* = 0.48) (Figure 2). The iCPC patients had slightly better long-term OS, although less than 25% at 5 years, indicating that the higher percentage of CPCs is associated with a biologically aggressive disease. In the multivariate Cox regression analysis, male sex, age, high-risk cytogenetics, high LDH, low platelets, and extramedullary disease were associated with inferior median OS (Table 3). There was no difference in the OS for Black vs. non-Black patients (HR 1.25, 95% CI 0.83–1.88; *p* = 0.28); similarly, the degree of leukocytosis had no impact on the OS (HR 1.34, 95% CI 0.90–2.01; *p* = 0.15). 

## 4. Discussion

We present a large single-institution body of data on PCL’s clinical, cytogenetic, and radiologic presentation and assess the impact of various therapeutic interventions on patients’ overall survival. The study included 54% Black patients (68% of female patients were Black), providing a unique opportunity to evaluate the clinical features and outcomes in a racially diverse, typically underreported population [20]. Although studies have shown that Black patients with MM have higher OS compared to White patients when receiving similar therapy, this was not extrapolated to the PCL setting; in our cohort, there was no OS difference between Black and non-Black patients [21].

The median age in our study cohort was younger than that reported for PCL in the literature, with a median age of 61 years, reflecting the large number of Black patients who present with MM at a younger age [22]. The median latency time between MM diagnosis and diagnosis of sPCL was short at 30 months, which is similar to other studies and probably reflects the presence of aggressive refractory clones at diagnosis [23]. Males were less represented in our study at 45%, which is an observation previously reported at our center for MM patients [24]. The laboratory features of our cohort mirror previously reported values, with patients with both pPCL and sPCL who tended to be anemic, thrombocytopenic, with abnormal renal function, elevated LDH and B2M, and several adverse features such as lytic lesions, extramedullary disease, and CNS involvement [10,25]. Interestingly, compared to sPCL and iCPC, the patients with pPCL in our study had higher calcium levels at diagnosis but were less likely to have lytic lesions on skeletal survey or PET/CT scan, suggesting systemic pathophysiology over localized osteoclastic bone resorption; hypercalcemia in PCL in the setting of tumor lysis syndrome has been reported [7]. Also, higher levels of myeloma-derived cytokines that activate osteoclasts may be a factor. Compared to pPCL, patients with sPCL and iCPC may be more likely to be exposed to bisphosphonates, which suppress hypercalcemia via bone remodeling and also have a longer time from diagnosis of MM to detection of CPCs associated with better control of lytic lesions and hypercalcemia.

High-risk cytogenetics were seen in >90% of the pPCL and sPCL cases, with >61% having a complex karyotype. Deletion 17p was also noted with a higher incidence in the sPCL (66%) and iCPC (50%) groups. In contrast to previously reported higher incidence of t(11;14) in up to 52% of pPCL patients, t(11;14) was only seen in 19% of our pPCL cohort [26]. Moreover, t(11;14) was also interestingly uncommon in Black patients in our cohort, presenting in 18% of the total Black cohort (n = 9), 15% pPCL, 24% sPCL, and 11% iCPC. 

The OS remains poor, even with combination regimens including daratumumab, IMiDs, PIs, and cytotoxic chemotherapy [27]. Of note, the DVT-PACE regimen, which is usually promoted as an upfront therapy, especially in pPCL, showed no OS benefit in our cohort [28]. This is probably secondary to the selection bias rather than the systemic assessment of the regimen and is likely explained by the introduction of DVT-PACE in a refractory setting after the failure of upfront induction, given the chemo-refractory nature of this population and with deletion of 17p detected in more than half of the patients. The OS in this study is similar to what has been reported from a recent multicenter retrospective study of 150 patients with a median OS 12.6 months [29].

In our patients with pPCL, induction with current regimens, including IMIDs, proteosome inhibitors, and daratumumab followed by ASCT led to marginally improved OS (HR 0.54, 95% CI 0.30–0.99; *p* = 0.05) compared to the sPCL patients, who had dismal OS. A retrospective multicenter study by the Greek Myeloma Study Group investigated patients with pPCL defined as ≥5% CPCs. They reported that patients who received VRd (bortezomib-lenalidomide-dexamethasone) or daratumumab-based quadruplets had a higher complete response (CR) rate of 41%, with an improved median PFS of 25 months and a median OS not reached in comparison to conventional chemotherapy or best supportive care [30]. Other groups reported conflicting data regarding the magnitude of the improved outcomes with aggressive regimens, including allogeneic transplantation [18,31,32]. Anti-B-cell maturation antigen (BCMA) CAR-T cell therapy has shown efficacy in pPCL in a phase 1 trial, with one patient achieving CR with a PFS of 307 days and another achieving a very good partial response (VGPR) with a PFS of 117 days [33]. Treatment of pPCL and sPCL remains challenging and the outcomes remain poor, in part due to the aggressive presentation and the refractory nature of these disease states. This is further complicated by the exclusion from clinical trials, thus limiting the systemic prospective assessment of various therapies. The change in diagnostic criteria from ≥20% CPCs to ≥5% CPCs for pPCL by the IMWG is a step in the right direction to ameliorate this issue. 

This study is limited by its retrospective, single-center nature. Although we enrolled 104 patients, the total numbers of patients in each group were relatively small (pPCL, sPCL, iCPC). The iCPC group (n = 20) probably represents the initial stage of development of PCL and assessment of the CPCs during routine follow-up may allow for early intervention. The treatment regimens were analyzed by drug class and not specific regimens (triplets, quadruplets), and they were subject to clinician’s bias and the nature of relapse. We focused on OS as an actual response and the PFS data were difficult to assess in terms of the impact of each therapy, as most patients were on continuous therapy and many suffered rapid relapses. Despite these limitations, our study provides further insight into an uncommon presentation of MM. 

## 5. Conclusions

To the best of our knowledge, this is the largest study from a single institution to evaluate PCL, with one of the highest inclusions of Black patients (54% overall, 68% of females). We demonstrated that salvage cytotoxic chemotherapy, including DVT-PACE regimens, despite being commonly used, provides no OS benefit, with the few responding patients who proceeded with SCT having the most benefit. The results of our study emphasize the need for a better understanding of the biological drivers of CPCs, especially at lower numbers, the optimal cut-off for the definition of PCL and its subgroups. There is an urgent need to investigate current immunotherapies such as Cereblon modulators, CAR-T, antibody–drug conjugates, and bispecifics, possibly in combination, to provide a meaningful impact for PCL patients.

## Figures and Tables

**Figure 1 cancers-16-02149-f001:**
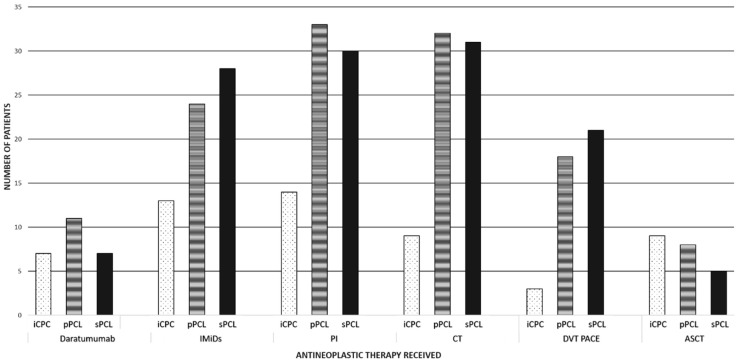
Distribution of antineoplastic therapy by group: iCPC (*N* = 20), pPCL (*N* = 35), sPCL (*N* = 49). IMiDs: immunomodulatory drugs (lenalidomide, pomalidomide, or thalidomide), PI: proteasome inhibitors (bortezomib or carfilzomib), CT: chemotherapy (cyclophosphamide, bendamustine, melphalan, cisplatin, etoposide, doxorubicin), DVT-PACE: dexamethasone, bortezomib, thalidomide, cisplatin, doxorubicin, cyclophosphamide, etoposide), ASCT: autologous stem cell transplantation. iCPC: intermediate circulating plasma cell (1–4% CPCs); pPCL: primary plasma cell leukemia, sPCL: secondary plasma cell leukemia.

**Figure 2 cancers-16-02149-f002:**
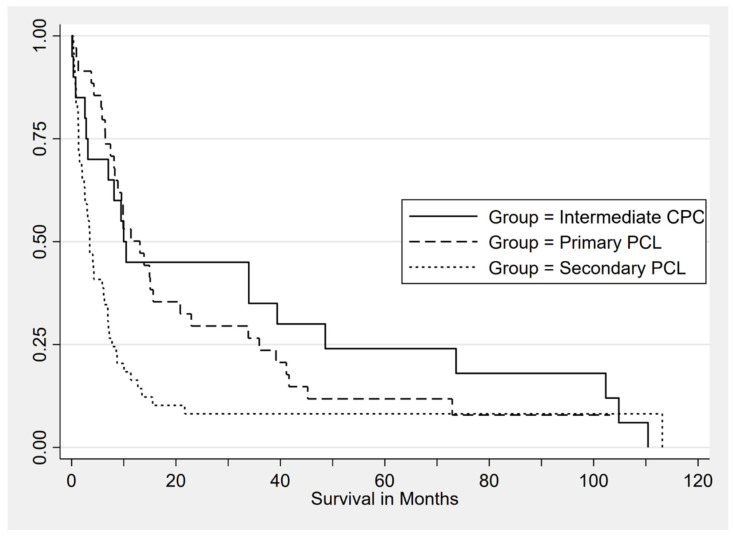
Kaplan–Meier estimate of overall survival by group: iCPC (*N* = 20), pPCL (*N* = 35), and sPCL (*N* = 49).

**Table 1 cancers-16-02149-t001:** Patients’ description and/or characteristics at diagnosis (*N* = 104).

	1–4% CPCs (iCPC) [*N* = 20, (%)]	Primary PCL (pPCL) [*N* = 35, (%)]	Secondary PCL (sPCL) [*N* = 49, (%)]	*p* Value
Demographics				
Age: years (mean, range)	58 (47–78)	57 (25–84)	58 (27–77)	0.82
Sex: male	9 (45%)	17 (49%)	21 (43%)	0.87
Race: Black	9 (45%)	23 (66%)	24 (49%)	0.21
Laboratory (mean, range)				
White blood cell count (×10^9^/L)	7.2 (1.4–36.8)	23.4 (1.2–90.6)	10.7 (0.5–49.6)	<0.001
Hemoglobin (g/dL)	9.3 (6.7–11.5)	8.2 (4.9–13.7)	8.2 (3.9–13.1)	0.03
Platelet (×10^9^/L)	75.6 (7–231)	101.2 (17–232)	61.7 (5–376)	0.03
% CPCs on CBC by morphology	1.9% (1–4%)	39.8% (2–95%)	30.3% (3–95%)	<0.001
Lactate dehydrogenase (units/L)	975.9 (175–5249)	918.7 (160–8348)	1085.1 (154–4579)	0.82
Creatinine (mg/dL)	1.4 (0.5–4.4)	2.7 (0.7–13.6)	2.2 (0.6–10.9)	0.18
Calcium (mg/dL)	9.1 (7.5–10.9)	9.9 (6.4–13.0)	9.3 (6.8–14.1)	0.02
Albumin (g/dL)	3.2 (2.0–4.4)	3.6 (2.0–5.1)	3.4 (1.8–6.7)	0.18
Beta-2 microglobulin (mg/L)	5.4 (1.6–25.7)	10.7 (1.1–74.2)	15.5 (1.7–169.1)	0.25
IgG subtype	9 (45%)	20 (57%)	30 (61%)	0.47
IgA subtype	7 (35%)	4 (11%)	10 (20%)	0.11
LC only	10 (50%)	18 (51%)	31 (63%)	0.44
Non-secretory	2 (10%)	3 (9%)	6 (12%)	0.86
R-ISS II; III	0 (0%); 20 (100%)	0 (0%); 34 (100%)	1 (2%), 46 (98%)	0.56
BM plasmacytosis (mean, range)	62.8% (10–95%)	78.1% (20–100%)	69.4% (3–95%)	0.22
Imaging				
Lytic lesions (skeletal survey; PET)	17 (94%); 19 (95%)	13 (48%); 21 (60%)	29 (73%); 34 (79%)	0.004; 0.01
Extramedullary disease (PET)	10 (50%)	14 (40%)	25 (51%)	0.58
Lymphadenopathy	2 (11%)	7 (23%)	11 (26%)	0.39
CNS involvement (MRI and/or LP)	1 (5%)	2 (6%)	7 (14%)	0.31
Cytogenetics				
Complex karyotype	4 (21%)	26 (79%)	30 (65%)	<0.001
Hypodiploidy	1 (6%)	14 (47%)	15 (33%)	0.01
Hyperdiploidy	8 (44%)	12 (40%)	17 (37%)	0.86
t(11;14); t(4;14); t(14;16); t(14;20)	2 (11%); 1 (6%); 0 (0%); 1 (6%)	6 (19%); 5 (16%); 7 (22%); 0 (0%)	8 (18%); 7 (16%); 6 (14%); 0 (0%)	NS
Gain 1q21; amp 1q21; del 1p	10 (59%); 3 (17%); 0 (0%)	22 (69%); 13 (39%); 10 (31%)	35 (80%); 18 (41%); 11 (26%)	0.24; 0.17; 0.03
Monosomy 13	5 (28%)	22 (65%)	23 (51%)	0.04
Del 17p	9 (50%)	10 (29%)	29 (66%)	0.006
Del 16q	0 (0%)	5 (15%)	2 (5%)	0.09
*MYC* rearrangement	0 (0%)	0 (0%)	3 (7%)	0.17
Flow Cytometry				
CD38	20 (100%)	32 (94%)	44 (92%)	0.70
CD138	20 (100%)	46 (96%)	29 (88%)	0.35
CD56	7 (50%)	8 (25%)	20 (46%)	0.27
CD19	1 (8%)	1 (3%)	1 (2%)	0.66
CD20	1 (7%)	3 (9%)	2 (5%)	0.04
CD45	3 (23%)	6 (21%)	9 (24%)	0.62
CD117	4 (29%)	6 (20%)	7 (18%)	0.15
CD52	0 (0%)	0 (0%)	1 (5%)	0.83
CD71	0 (0%)	1 (33%)	1 (25%)	0.80

Abbreviations: PCL, plasma cell leukemia; CPC = circulating plasma cell; iCPC = intermediate CPC; *N* = number; R-ISS: Revised-International Staging System; BM = bone marrow; PET = position emission tomography; CNS = central nervous system; MRI = magnetic resonance imaging; LP = lumbar puncture; NS = non-significant. No patients in this study were of the IgM subtype.

**Table 2 cancers-16-02149-t002:** Antineoplastic therapy effect on the overall survival in all patients (*N* = 104).

Antineoplastic Therapy	Hazard Ratio (95% CI)	*p* Value
Daratumumab	0.43 (0.26–0.71)	0.001
IMiDs	0.13 (0.08–0.22)	<0.001
PI	0.35 (0.22–0.56)	<0.001
Chemotherapy	0.69 (0.44–1.08)	0.11
DVT-PACE	0.97 (0.62–1.50)	0.88
ASCT	0.54 (0.30–0.99)	0.05

IMiDs: immunomodulatory drugs (lenalidomide, pomalidomide, or thalidomide), PI: proteasome inhibitors (bortezomib or carfilzomib), CT: chemotherapy (cyclophosphamide, bendamustine, melphalan, cisplatin, etoposide, doxorubicin), DVT-PACE: dexamethasone, bortezomib, thalidomide, cisplatin, doxorubicin, cyclophosphamide, etoposide), ASCT: autologous stem cell transplantation.

**Table 3 cancers-16-02149-t003:** Results of the multivariate Cox regression models for the overall survival in all patients (*N* = 104).

Variables	Hazard Ratio (95% CI)	*p* Value
Male Gender	1.5 (1.0–2.3)	0.04
Older Age (>58 years old)	1.7 (1.1–2.5)	0.02
High-Risk Cytogenetics	2.5 (1.3–5.0)	0.01
High LDH (>3× normal)	1.6 (1.0–2.4)	0.04
Low Platelets (<50 × 10^9^/L)	1.8 (1.2–2.7)	0.006
Extramedullary Plasmacytoma (PET Scan)	1.6 (1.0–2.4)	0.03

Results from the entire cohort (n = 104). LDH, lactate dehydrogenase; PET, positron emission tomography. High-risk cytogenetics (TP-53), t(4;14), or t(14;16).

## Data Availability

The data supporting the findings of this study are available upon request from the corresponding author.

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
