# Peer review of "A Clinical Perspective on Plasma Cell Leukemia: A Single-Center Experience"

_cancers, 2024, doi:10.3390/cancers16112149_

Round 1

Reviewer 1 Report

Comments and Suggestions for Authors

In the present study Li et al, present their single-center experience on the clinical management of PCL patients (pPCL and sPCL) by performing a retrospective analysis of patients cohort. 

Indeed this is a large study form a single-site including an adeuate number of pPCL and sPCL patients. Moreover, it has some interest the inclusion of the group with 1-4% CPCs as an intermediate group between MM and pPCL with probable high-risk features (in comparison with the majority of MM patients having low CPC levels).

Besides the limitations derived from the small numbers in each group and the retrospective nature of the study (which are also recognozed by the authors), the major problem is tha lack of particular novelty. 

The major finding of the study is that sPCL has a worse OS compared to pPCL and that the outcome of patients in the iCPC group have similar OS with that of pPCL. As of theapy, patients receiving chemotherapy or DVT-PACE did not have a clinical benefit compared with the Pi, IMiD, Daratumumab and ASCT groups, which is not something new.  

Other comments

- When I see at the patients' characteristics in Table 1, I notice some interesting  discrepancies compared with reported findings in the literature, which may have created some bias in patients' selection process. 

In particular: i) I see that almost all patients are of R-ISS III, which is not the case for patients with high CTC levels or even PCL; cytogenetics: the prevalence of t(11;14) is particularly low for pPCL (commented by the authors), on the other hand t(14;16) is high. Moreover, complex karyotype and hyperdiploidy is similar or more often in pPCL than sPCL which is not highly expected. iCPC group had more del17p than pPCL.

- In Table 3 with the effect of different therapies on OS, authors should explain which was the reference group in order to get these hazard ratios in each patient subgroup. 

Author Response

We thank the reviewer for his comments, we agree the diagnosis of PCL carries a poor prognosis as has been reported in the literature.

The discrepancy of presenting features of our patients, table 1, probably reflects a biological difference for black vs Caucasian patients. To our knowledge this is the first PCL study that provide larger number of black patients.  

RISS III is related to high B2M and high-risk cytogenetics which is a major feature of our patients. 

Our cytogenetic profile is interesting, we are seeing more del 17 and t(14:16) which we believe is the highest cytogenetic risk. While t(11:14) is not as high as usually reported the reasons are unclear and can be attributed to racial differences, though data is not conclusive.  

The HR in table 3 reflects the impact of therapy on the subgroups and mostly measuring the impact of each drug on OS.  

Reviewer 2 Report

Comments and Suggestions for Authors

This is an interesting manuscript that deals with a single-site experience with a rare but grave disease, plasma cell leukemia. It is a single center retrospective experience that encompasses two decades - this means that therapeutic drug availability significantly changed throughout the study period. This makes proper respone analysis a tough one. Nevertheless, the authors present a commendable attempt to reach one.

Notwithstanding the merits of such an analysis, I have some problems with the data that need clarification.

In Table 1, relavant lab results are presented, however, normal ranges for the lab values are not given, therefore it is not possible to judge e.g. levels of LDH. Moreover, the cutoff levels for positivity in the cytogenetic results are not presented. Additionally, I find the percentage of t(11;14) translocated PCL patients rather low: this may be due to the large black patient ratio, a unique feature of this study, anyway, needs explanation.

I was also surprized, that venetoclax therapy was not attempted in any of these patients, as this drug may show high activity in PCL that exhibits translocation t(11,14), especially if CD20+. Additionally, it is perplexing that no attempt was towards an allograft in this disease, for two deacades?

Actual protocols, and not individual drugs should also be attempted to get analyzed, only DVT-PACE results are presented in a clearcut manner.

Author Response

we thank the reviewer for these comments...

It is single center retrospective experience that encompasses two decades - and while it is true that the therapeutic landscape in MM has changed significantly, it is quite sobering to see no impact on OS in ultrahigh risk disease such as PCL, which remains an unmet need in MM.

we added normal ranges for LDH,

Cytogenetic cut off for FISH was not standardized and was reported when the lab reported positive values, while high cut off (e.g. 70% for del 17p) has been associated with worse outcome compared to lower cutoffs in PCL we did not find levels of positivity to be significant. 

The low levels of t(11:14) is quite interesting and the reviewer point about racial differences may be a factor though difficult to ascertain from our small numbers. 

allo SCT was done in 6 patients as mentioned in results and CAR T in 3 but the numbers was very low to analyze separately. And really has no impact on OS.

Venetocalx was not given in this cohort, we tried in few patients to get insurance approvals and progression was the main reason not adminster the drug. In our experience and in newer cohorts we added venetoclax to other combinations (carfilzomib and Daratumumab) and in 2 patients (not included in the current study) we noted brief minimal responses and rapid progression. A trial addressing this question is worth considering. 

While different regimens would be ideal, all patients received combinations of drugs and many were added to existing drugs due to lack of response, none of the combinations has any significant impact on OS. we selected DVT PACE (considered standard of care for PCL by many thought leaders); and again in our study no impact on OS just temporary responses with rapid progression. 

Reviewer 3 Report

Comments and Suggestions for Authors

First, I would like to thank the author for this fruitful work. This single-center retrospective study of Plasma cell leukemia provided a concise background and clinical outcome on the three categories of these rare diseases

1. The title is informative and representative of the study

2. The simple summary doesn't include any information about the CPC cohort.

3. The Abstract section is well-written and summarizes the manuscript.

4. The introduction section is well-written with a good presentation of the importance of the study. 

5. The study design is good but not novel.

6. In the result section 

a. some laboratory data are missing like urinary immunofixation and serum-free light chain assay if you have them please add a short note about them  

b. Line 100 the p- values don't match the table please re-check them 

c. The OS is linked to the class of drugs, not to the therapeutic regimen, and this approach creates bias. 

7. The figures and the tables are clear and descriptive 

8. Most of the references are updated and relevant to the study 

Overall the manuscript is informative and deserves to be published as it describes a single-center experience in a rare disease and highlights the importance of determining the level of CPC in MM and its clinical importance as those patients  should be monitored carefully and treated in a personalized way 

Author Response

we thank the reviewer for his positive comments about the study. 

These values adds little to PCL diagnosis and while available we did not include them to simplify the presented data.

P values and tables are checked and corrected.  

The OS is linked to the class of drugs, not to the therapeutic regimen, mostly as all patients received various combinations making it difficult to provide a specific regimen. The exception was DVTPACE which is considered a Standard of care in PCL by many and in our experience despite initial responses there was no impact ion OS. 

Round 2

Reviewer 1 Report

Comments and Suggestions for Authors

As I mentioned in my first report my main concern of this study is the lack of particular novelty. 

The study may include some increased number of black patients, but overall there are no novel findings compared to what has been already reported in the literature